# Health inequity: Possibilities of initiating pulmonary telerehabilitation programs for adults with chronic obstructive pulmonary disorders in conflict and low-resourced areas; A mixed-method phenomenological study

**Suad J. Ghaben**[1,2]*, **Arimi Fitri Mat Ludin**[3], **Badr Elkholi**[2,4], **Reem Kullab**[4], **Majd Al-Hour**[4], **Devinder Kaur Ajit Singh**[1]*

**1** Physiotherapy Programme & Center for Healthy Ageing & Wellness (H-CARE), Faculty of Health Sciences, Universiti Kebangsaan Malaysia, Jalan Raja Muda Abdul Aziz, Kuala Lumpur, Malaysia, **2** Department of Physiotherapy, Faculty of Applied Medical Science, Al Azhar University-Gaza, Gaza, Palestine, **3** Biomedical Science Programme & Center for Healthy Ageing and Wellness (H-CARE), Faculty of Health Sciences, Universiti Kebangsaan Malaysia, Jalan Raja Muda Abdul Aziz, Kuala Lumpur, Malaysia, **4** Physiotherapy department, Ministry of Health, Gaza, Palestine

* devinder@ukm.edu.my (DKAS); s.ghaben@alazhar.edu.ps (SG)

## Abstract

### Objective

The "triple problems of COPD"—underdiagnosis, underrecognition, and underdevelopment of pulmonary rehabilitation programs—are global. Pulmonary telerehabilitation (PTR) may solve these problems. This study aims to explore the possibility of launching PTR programs for adults in an area of conflict and low resources.

### Methods

A mixed methods convergent design was used to understand outpatient Pulmonary Rehabilitation (PR) unavailability. The medical records of 70 patients with COPD in the Ministry of Health (MoH) chest unit were analysed quantitatively. thirteen patients who received in-hospital physiotherapy were questioned via a structured and validated questionnaire, and 12 healthcare professionals (HCPs) were interviewed. Quantitative data was analyzed descriptively, and inferentially by the SPSS, outlining the means, standard deviations and percentages, and the chi-square, respectively. Qualitative data was analyzed thematically using the NVivo software, and five themes and 25 subthemes were identified. A thematic framework and triangulation model were depicted accordingly.

### Results

Seventy patients with a mean age of $65.3 \pm 1.65$ years and a mean hospital stay of $4.89 \pm 0.392$ days were managed with no admission, Physiotherapy (PT) referral, or

**Data availability statement:** All relevant data are within the paper and its Supporting Information files.

**Funding:** The author(s) received no specific funding for this work.

**Competing interests:** The authors have declared that no competing interests exist.

discharge criteria. Patients reported needing outpatient PR (4.38 ± 1.193) and a willingness to adopt (3.77 ± .832). Interviews with HCP highlighted the environmental and professional challenges, opportunities, and PTR traits. Quantitative and qualitative data triangulation led to the COPD Sequential Triad (CST) theory.

## Conclusion

This study identified the national, institutional, and professional challenges and opportunities of launching PTR programs in conflict and low-resourced areas, including the conflict's consequences, insufficient resources, and absence of a structured healthcare system and clinical practice guidelines. Promoting health equity in such areas necessitates that the international community and institutions isolate the healthcare sector from conflicts and apply more effective strategies. The study also illustrated the characteristics of the intended PTR programs, including mHealth, therapeutic devices, and hybrid programs along with supporting professional organizations' guidelines.

## Introduction

Chronic Obstructive Pulmonary Disorder (COPD) is the third and sixth leading cause of death [1] and disability [2] worldwide, affecting 4% of the population, with higher prevalence and mortality in Lower- and Middle-Income Countries (LMICs) [2,3]. The impact of conflicts on health communities leads to a high incidence of injuries and a lack of disease prevention and health promotion for noncommunicable diseases such as COPD. It also jeopardises the social determinants of health (SDH) and the population's well-being. Conflict leads to shortages in the healthcare system in terms of diminished quality of care, restricted access, and inadequacy of resources and experience [4,5]. Crucially, the conflict in Palestine has damaged the health infrastructure to an extent that deprived citizens of basic healthcare [6]. Palestine is an occupied territory and a conflict area, and ranks third and fourth in the COPD burden in the West Bank and Gaza Strip, accounting for 12.2% and 12.6% of Disability-Adjusted Life Years (DALYs) [7]. COPD is the fourth cause of crude mortality in Gaza at 10.4% and the third of years of life lost (YLLs) at 3.1 [7]. The latest COPD burden for the MENA region highlighted that the age-standardized prevalence of COPD per 100,000 population was 2000 to < 2500, the number of cases was 11.42 (10.3 to 12.69) million, and the DALYs was 2.53 (2.31 to 2.8) million cases [8]. Despite the growing economic and social burden of COPD [1], it remains a low priority in healthcare policy worldwide [9], particularly in areas of low resources, [10,11], and affected by conflict and violence [7].

Pulmonary Rehabilitation (PR) in High-Income Countries (HICs) faces multiple challenges, including low referrals, uptake, and adherence [9], and cultural sensitivity affecting PR uptake [12]. PR in LMICs are either unavailable [13] or have restricted access [9], restricted stakeholder engagement [11], and unorganised delivery [10]. Environmental factors such as the long travel distances and time, use of public

transportation, weather, waiting time, and inappropriate health system resources limit access to appropriate programs [14,15]. Personal factors, including health illiteracy [9,12] and risk factors, impede PR completion [14,16,17]. Lack of Health Care Professionals' (HCPs) skills and referral standards also hampers PR provision [14]. The "triple problems of COPD": underdiagnosed cases [18], under-recognized disease burden [7,9–11], and underdeveloped PR programs [7,13], indicate the need for sustainable client-centred PR such as Pulmonary Telerehabilitation (PTR) with technology-based adjuncts. American Thoracic Society (ATS) and European Respiratory Society (ERS) recommended PTR to provide sustainable PR solutions [19]. Others The Forum of International Respiratory Societies [2], and other investigators claimed that facilitating access to spirometer detected undiagnosed cases in HICs and LMICs [2,18,20].

PTR is effective in managing COPD symptoms [21]. It enhances PR program adherence by increasing accessibility [22,23] and facilitating deployment with limited resources [24]. Remote home monitoring improves treatment outcomes and patient and HCP satisfaction, with HCPs' feedback decreasing hospitalisations [25]. TR provides screening, evaluation, supervised intervention, self-management, psychological support, monitoring, and follow-up via mobile or online apps, hardware, and the cloud. While having HCPs actively supervise treatment regimens, assess patient progress, or coordinate care [26]. Interactive features or gaming could improve peer support, modelling, and motivation [22]. The Smart-OPEP, which was proposed as a form of PTR, integrates the Oscillatory Positive Expiratory Pressure Devices (OPEP) with a spirometer and mobile app and is intended to solve the unavailable PR and enhance OPEP functionalities [27]. The OPEP improves physiologic and functional outcomes of COPD and may enhance Quality of Life (QoL) [27,28]. Spirometer measurements confirm COPD diagnosis [1] to guide emergency department and hospital admission and discharge [29]. It also guides the one-month early discharge [30], predicts late post-discharge relapse [31], and strongly predicts mortality and readmission [32]. Spirometers are still underutilised in COPD diagnosis and management [33]. To enhance their utilisation, easy-access portable spirometers were developed [33–36].

Quality care and innovative services require a structured healthcare system [37] with health policy, clinical practice guidelines, health workforce, pharmaceuticals, medical goods, health information systems reporting, governance, and financing [38]. The healthcare system in the occupied territory is fragmented, under-resourced, and technologically outdated; its services are provided by the Ministry of Health (MoH), the United Nations Relief and Works Agency (UNRWA), the private sector, and international Nongovernmental Organizations (INGOs) [39]. Palestinian healthcare system lacks financing, good governance, evidence-based policies, information, and database accessibility, and sector coordination [40]. Current restrictions limit medical supplies, equipment, medications, experts, and patients' movement across the country. During attacks, some healthcare facilities were destroyed and shut down, limiting the MoH's control over land, water, and energy and impeding its ability to provide healthcare [39,41–43]. The lack of outpatient PR in the target MoH prevents patients with COPD from receiving a continuum of care. PTR with technology-based adjuncts may help sustain PR and address healthcare system shortages. This research was conducted to identify [1] challenges, [2] opportunities, and [3] characteristics of initiating PTR programs with technology-based adjuncts.

## Materials and methods

### Study design

This mixed-methods study integrated quantitative and qualitative research and pragmatism [44,45], to generate a comprehensive understanding of the phenomena [46,47] related to the unavailability of outpatient PR programs in conflict low-resourced areas, and the possibility of launching alternative PTR programs. We collected quantitative data using document analysis and a questionnaire and qualitative data using in-depth interviews (IDI), as shown in Fig 1. This formative research will inform the Smart-OPEP development [27]. Using purposive sampling, we adopted the descriptive, cross-sectional design for the quantitative part and the phenomenological method for the qualitative part. A local MoH committee (993123) approved the research protocol. All participants were informed about the study's purpose, data collection

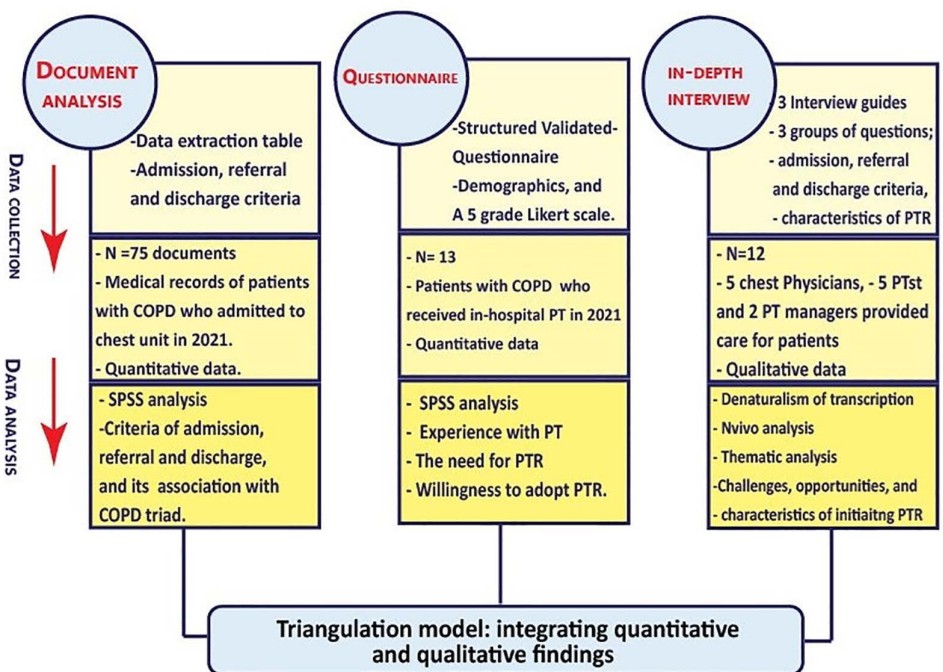

**Fig 1. Applying triangulation (convergent parallel MMR) to identify the possibilities of launching PTR.**

process, and telerehabilitation intervention. After obtaining informed consent, all questionnaires and IDI were performed through face-to-face individual interviews.

## Data collection

The three study instruments, data extraction table, questionnaire, and interview guides, were prepared after identifying each tool's goal and target population. From 23rd January to 30th January 2022, SG concurrently conducted a preliminary interview with a chest doctor and a physiotherapist to identify their experience with patients diagnosed with COPD and explored randomly selected medical records. Based on the preliminary interview results, data from medical records, and literature search, SG prepared the draft version of the three study tools, which were discussed thoroughly among the team (SG, BKH, RK, MH, AFML, and DKAS) and further refined. The questionnaire was developed from the draft prepared by SG. In addition to the SD section, the questionnaire measured the patients' experience with in-hospital PT, the need for outpatient PR, and the willingness to adopt PTR. Before administration, the questionnaire was validated for face and content validity by three physiotherapy researchers.

## Quantitative data

Quantitative data was collected through document analysis and questionnaires.

## Document analysis

We analysed medical records [48] of patients with COPD admitted to the MoH chest unit from January 1 until December 31, 2021. We applied the management framework as described by GOLD [1]. Data accession and extraction were performed from 21st February to 9th March 2022 by RK and MH utilising the data extraction table. We acquired permission

before accessing medical records. SG reviewed and verified the data from the extraction table. The table encompassed primary data (gender, age, and length of hospital stays) and seven sections, each with a drop-down checklist: [1] admission criteria; [2] investigations; [3] referral to Physiotherapy (PT); [4] medical management; [5] progress notes; [6] comorbidities; and [7] discharge criteria. The MoH information system allows researchers to access information that identifies individual participants during data collection.

## Questionnaire

Data was collected from 7th to 9th March 2022 by RK and MH using the structured and validated questionnaire in an interview context. All participants provided written informed consent. Patients with COPD admitted to the chest unit in 2021, who received in-hospital PT and were discharged from the hospital in 2021, regardless of the number of days post-discharge, were included. The questionnaire was developed in Arabic from the draft prepared by SG and according to the preliminary interviews and the medical records. The Arabic questionnaire was translated into English for review and verification by AFML and DKAS. The questionnaire measured the patients' experience with in-hospital physiotherapy, their need for post-discharge PR, and their willingness to adopt PTR. It had two sections: the first section comprises the demographic data: smoking, comorbidities, medical history, COPD diagnosis date, symptoms, and intervention received post-discharge. While the second section encompasses three subsections with a total of 27 items and measured agreement on a five-point Likert scale, inquiring about [1] inpatient Physiotherapy (PT) management experience, [2] the need for PTR post-hospitalisation, and [3] willingness to receive pulmonary telerehabilitation. Each subsection was followed by a general inquiry scale, rated 1–5, to asses patients' satisfaction with inpatient PT, the need for PTR, and telerehabilitation willingness. The Arabic questionnaire's alpha Cronbach for the subsections was 0.985, 0.973, and 0.879, indicating high reliability. Content validity (CV) results demonstrated high validity; the I-CVI for the 27 items and the S-CVI scored 1 and were calculated using data from three evaluators.

## Qualitative data

We collected qualitative data for this study by conducting 12 IDIs from 27th February to 17th March 2022. We interviewed five chest physicians and five physiotherapists who treated patients with COPD. We also interviewed two PT managers who made clinical and administrative decisions, including the head of the Physiotherapy department at the hospital and the general manager of the Physiotherapy and Rehabilitation unit at the MoH. Interviews were conducted in the MoH's participants' offices using interview guides and in Arabic, allowing participants to openly express their viewpoints. All three guide versions had three sections: [1] doctors-physiotherapists communication, [2] medical and PT management, and [3] PTR needs and acceptance. SG obtained the informed consent and moderated the interviews, and BKH took notes and audio-recorded them. BKH prearranged all interviews via phone and sent participants information sheets. Participants' trust and rapport, which were built through prolonged engagement with the research team, helped sustain credibility.

## Data analysis

**Quantitative data.** Medical records and questionnaires were analysed using IBM SPSS 22. Records about participant age, gender, hospital stay, medical investigation, management, comorbidities, and discharge criteria were analysed using descriptive statistics. Inferential statistics were deployed to analyse the correlation between referral to PT and dyspnea, coughing, and sputum production. Questionnaire data identified participants' gender, age, length of COPD diagnosis, comorbidities, last week's COPD symptoms, and hospital discharge care using descriptive statistics.

**Qualitative data.** Findings were reported according to the Consolidated Criteria for Reporting Qualitative Research (COREQ) [49]. MH and RK transcribed verbatim the Arabic recorded interviews, then translated them into English, considering the interpretation of the talking data [50]. SG and BKH, line by line, reviewed and verified the interview scripts.

SG used the NVivo 12 to categorise data into initial codes, merge and shift codes into categories, and produce themes from categories to answer the research question [51,52]. Codes and themes were phrases or full sentences that captured the essence of the extracts. SG discussed the themes with DKAS, AFML, and BKH until an agreement was reached. Member checking and triangulation were applied to verify the thematic analysis [53,54]: participants' trust and rapport were attained through prolonged engagement; researchers and HCPs were involved for years in varying projects [54].

**Triangulation.** A well-justified MMR design and suitable triangulation techniques are essential for credible and rigorous results [55,56]. This study used triangulation and reporting recommendations [55,57–59] to organise and integrate the qualitative and quantitative data, enrich the data, and verify the results.

## Results

The study results were derived from document analysis, questionnaires, and IDI. These results were triangulated in the Convergent parallel MMR triangulation model.

### Document analysis

**Participants.** Document analysis and demographics of the questionnaire are shown in S1 Table.

The study comprised 70 medical records of patients with COPD hospitalised in the chest unit. For two females and 68 males, their average age and hospital stay were $65.3 \pm 1.65$ years and $4.89 \pm 0.392$ days, respectively.

The top five criteria for admitting patients to the chest unit were SPO2, dyspnea, coughing, respiratory rate and high temperature, recorded in 90%, 75.7%, 71.4%, 51.4%, and 42.8% of the records, respectively. In 91.4%, 75.7%, 72.8%, 72.8%, and 71.4% of records, CBC, ECG, electrolytes, ABGs, and chest X-ray were the top five investigation procedures. Spirometry and bronchoscope were never used for investigation. Although admission and investigation outcomes were consistent, they did not follow any guidelines. The lack of consistently validated symptom scales for COPD diagnosis and GOLD guidelines was evident. The most commonly prescribed medications were supplements, antibiotics, inhaled bron-chodilators, O2 therapy, inhaled antispasmodics and antibiotics (84.3%, 78.6%, 57.1%, 47.1%, 40%, and 32.8%, respectively). PT was inadequately represented in patients' records, with just 7.1% showing a written referral and 10% recording a verbal "discussion with physiotherapist". In contrast, 82.8% of medical records did not mention PT.

Patients usually have COPD and other comorbidities. Hypertension, congestive heart failure, type-2 diabetes, and smoking were the most prevalent comorbidities, with 45.7%, 32.9%, 31.4%, and 28.6%, respectively in medical records. In 51.4%, 30%, and 27.1% of the records, discharge decisions were based on increased SPO2, dyspnea alleviation, and chest clearance, respectively. Improved body temperature, heart rate, and respiration rate were recorded in 30%, 24.3%, and 20%, respectively of records. Only 21.4% and 1.4% of records listed air entry and lung function tests, respectively, the most critical indications of improved COPD.

The chi-square test examined the association between PT referral, the COPD triad, dyspnea, cough, and secretion production. About 68.6% of patients in the chest unit were not referred to PT, while 62.3%, 64%, and 70% (n = 70) of dyspnea, coughing, and sputum patients, respectively, were not referred to PT. The association between dyspnea and the likelihood that patients were being referred to PT was significant ($p < 0.05$), while it was insignificant ($p > 0.05$) for coughing and sputum production.

### Questionnaire

**Participants.** Thirteen patients with COPD were admitted to the chest unit and received in-hospital PT completed the questionnaire. The average age was $66,07 \pm 9.84$ years, BMI was $35.2 \pm 13.35$, and years since diagnosis with COPD were $6.88 \pm 7,52$ years. Most patients had low-level education, while 7.7% had a university degree, and 7.7% were employed.

Walking difficulties, stress and anxiety, diabetes, hypertension, cardiovascular disorders, and current smoking were reported by 76.9%, 61.5%, 53.8%, 46.2%, and 38.5%, respectively, of respondents. Dyspnea, fatigue, inability to conduct ADL, coughing, and sputum were reported by 92.3%, 84.6%, 84.6%, 69.2%, and 69.2% of participants, respectively. About 23.1% of participants continued to receive PT after discharge, mostly private sessions. In comparison, 100% used drugs to treat COPD, and 46.2% consulted a chest doctor.

The patients' experience with the in-hospital PT, their needs for outpatient PR, and their willingness to adopt PTR are illustrated in S2 Table. Regarding in-hospital PT, patients had moderate acceptance of care quality, with mean values ranging from 2.31±1.377 to 2.69±1.653 for specific services. They reported lower levels of satisfaction with care (2.15±1.573). Although post-discharge PT was not a routine practice in the hospital, it was considered valuable by patients, with mean values ranging from 3.77±1.481 to 4.54±1.19, and overall need scored a higher value of 4.38±1.193. Patients were willing to adopt telerehabilitation services, with mean values ranging from 3.23±1.013 to 4.31±1.032 for specific tools and an overall willingness of 3.77±.832. Patients who preferred home PT, either conventional or TR, scored highest, with mean values of 4.54±1.198 and 4.31±1.032, respectively.

### The in-depth interviews

**Participants.** Five chest physicians (average experience: 20±7.03 years) and five physiotherapists (average: 11±.7.74 years) were interviewed. The PT department director and the PT clinics' general directorate at MoH were also interviewed.

Three categories of possibilities for launching PTR in the target area were identified: challenges, opportunities, and characteristics of PTR. Fig 2 shows the thematic framework. The coded interview transcripts identified five themes: environmental challenges, professional challenges, environmental opportunities, and characteristics of PTR. Each theme was described into subthemes, with a total of 25 subthemes.

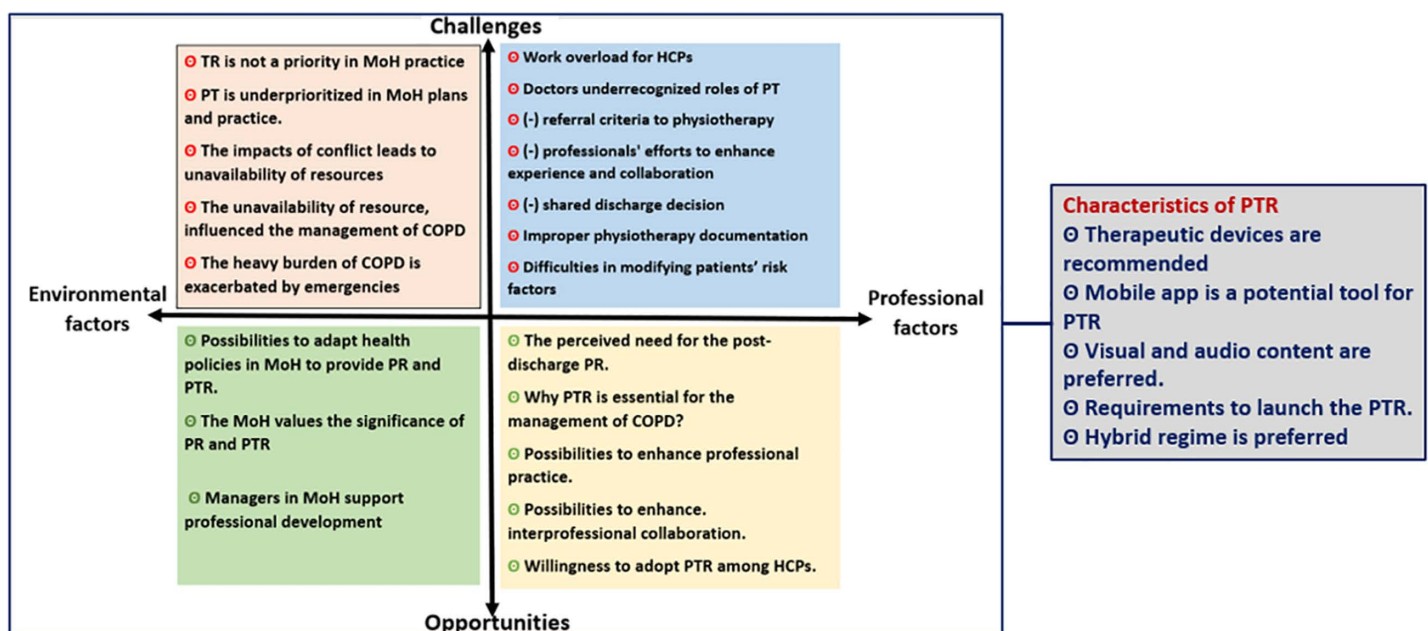

**Fig 2. Thematic framework. (-) indicates lack of. PT: Physiotherapy, PR: Pulmonary Rehabilitation, PTR: pulmonary telerehabilitation.**

Challenges were classified into environmental and professional challenges, with participants' numbers and sample quotations for each subtheme shown in S3 Table. Physiotherapists were referred to P1 to P5, doctors to D1 to D5, and PT managers to M1 and M2. Five subthemes described the environmental challenges: "Telerehabilitation is not a priority in the MoH practice", "Physiotherapy is under-prioritised in MoH plans and practice", "The impacts of conflict lead to unavailability of resources", "The unavailability of resources influenced the COPD management", and "The heavy burden of COPD management is exacerbated by emergencies." Seven subthemes described the professional challenge: "Work overload for doctors & physiotherapists", "Doctors underestimate the roles of PT", "Lack of referral criteria to PT", "Lack of professionals' efforts to enhance experience and collaboration", "Lack of shared discharge decision", "Improper PT documentation", and "Difficulties in modifying patients' risk factors."

Opportunities were classified into environmental and professional, as shown in S4 Table. Three subthemes described the environmental opportunity: "Possibilities to adapt health policies in MoH to provide PR and PTR services", "MoH values the significance of PR and PTR", and "Managers in MoH support professional development and collaboration." Five subthemes described the professional opportunities: "The perceived need for the post-discharge PR", "Why PTR is essential for the management of COPD?", "Possibilities to enhance professional practice", "possibilities to enhance interprofessional collaboration", and "Willingness to adopt PTR among HCPs."

Five subthemes described characteristics of the intended PTR, as shown in S5 Table: "Therapeutic devices are recommended", "Mobile app is a potential tool for PTR", "Visual and audio contents are preferred", "Requirements to launch the PTR", and "Hybrid PTR regime is preferred."

### The triangulation model

This convergent parallel study included results from records, questionnaires, and interviews to investigate the challenges and opportunities of launching PTR. The integrated results added to the literature on the "triple problems of COPD": underdiagnosis [18], disease under-recognition [7,9–11], and PTR underdevelopment [7,13]. The HCPs highlighted the underdiagnosis: "almost two-thirds are COPD but not diagnosed." Fig 3 depicts the disease under-recognition and PR underdevelopment according to the COPD Sequential Triad (CST).

At the national level, three factors influenced the triple COPD challenges: the conflict and its sequels, as stated by four HCPs: "the siege"; the low-resourced settings: "the unavailability of resources"; and the socioeconomic influence: "the heavy burden of COPD is exacerbated by emergencies" and "difficulties in modifying patients' risk factors."

Document analysis revealed a lack of institutional clinical practice guidelines and standardised admission criteria. Nine HCPs reported a "lack of referral criteria to PT," and document analysis supported that. Ten HCPs reported a "lack of shared discharge decision" "once the patient improved." Unconsolidated discharge criteria were supported by document analysis. The highest score was 50.4% for SPO2 in patient records. Although discharge planning was not mentioned in patients' records, the questionnaire showed that all discharged patients continued their medications, 46.2% continued physician follow-ups, and 23.1% continued PT. HCPs indicated, "I created a follow-up program with some patients" and "simply referred patients to the outpatient clinic."

The quality of in-hospital PT indicated a "lack of professionals' efforts to enhance experience and collaboration" by five HCPs, while patient satisfaction was poor (2.15±1.573). Eight HCPs recorded "The Perceived Need for Post-Discharge PR" for outpatient PR, and three recorded "The MoH, staff values the significance of PR and PTR." Patients emphasised this (4.38±1.193). Nine HCPs recorded the importance of PTR in COPD management, and patients indicated a willingness to adopt it (mean value of 3.77±.832). Additionally, eleven individuals indicated a "willingness to adopt PTR among HCPs."

Nine HCPs provided PTR characteristics, indicating willingness to use the mobile app and patient support (mean value of 3.23±1.301). Five HCPs and patients reported using OPEP as an additional treatment device (mean value: 3.69±1.182). Two HCPs chose the PTR program's visual and auditory components, and patients preferred learning therapeutic exercise through video (mean value=3.77±1.013).

 

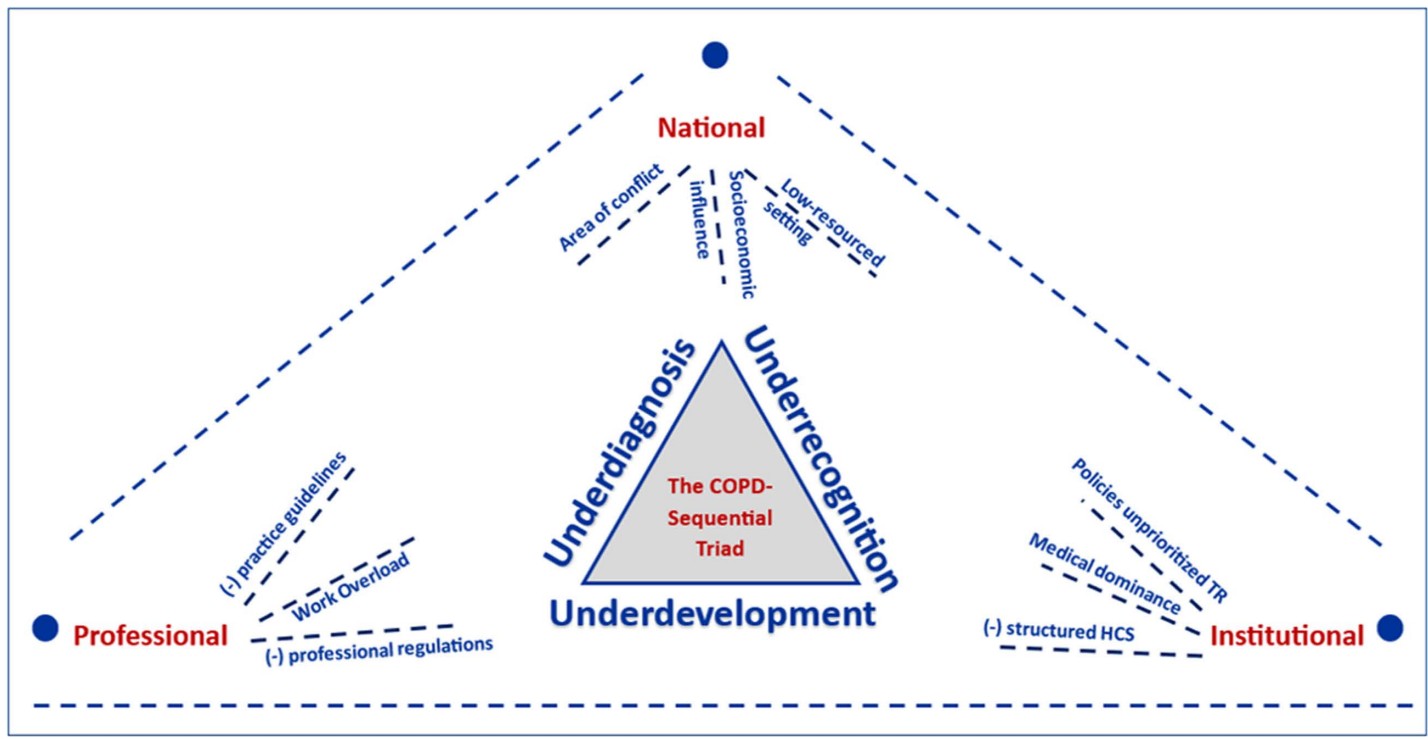

**Fig 3. The triangulation model represents the CST model based on the present study's findings. Three key factors primarily impact the triple problems associated with COPD. The conflict area, the low-resource setting, and the socioeconomic factors influencing the population serve as illustrations at the national level. [2] At the institutional level, there is a lack of a structured healthcare system, medical dominance, and policies that undervalue PR. The absence of clinical practice guidelines and professional regulations and the overloading issue contribute to the professional factor. (-) indicates lack of HCS: healthcare system.**

Ultimately, patients and HCPs prefer using the mHealth and therapeutic devices. Therefore, the Smart-OPEP, as a PTR tool, will include mHealth and therapeutic devices such as OPEP with a preference for video and audio inclusion of the mHealth. A hybrid regime is recommended as a start, and professional association guidelines should support the administration of PTR. It is worth noting that communication among patients and HCPs will be maintained through the mHealth app as a preferred PTR tool.

## Discussion

This study highlighted the environmental and professional challenges, opportunities, and characteristics of initiating PTR in a conflict and low-resource area. It also outlined the national, institutional, and professional factors that influence the initiation of PTR. This work also emphasized the possibilities of the transformation to telehealth even in conflict and low-resourced areas, which aligns with the fourth industrial revolution [60]. By tackling the identified challenges and seizing the available opportunities, such novelty can be achieved. Launching PTR programs also requires the allocation of resources, working in MDTs, and aligning with the appropriate frameworks and models [26,61].

Our analysis stated that the target area's conflict status and low resources hindered PTR implementation, impacting the country's socioeconomic status. Thus, resource constraints led the MoH to eliminate TR from current and future practice, which aligns with previous findings. [43,62] alarming the national, institutional, and humanitarian crises due to the impacts of conflict [6,41], and leading to the devastation of the MoH [6,43,62–64]. The impacts of conflict, and uneven distribution

of power, money, and resources among the healthcare system's pillars, led to lowering the QOL [62,64,65], which has greatly affected social determinants of health and health outcomes [66,67]. The WHO recommends improving daily living, addressing power, money, and resource inequalities, and monitoring and evaluating the problem and required action to improve health equity and outcomes [66]. The latter seems most feasible for researchers, while affecting daily life and power, money, and service inequalities are beyond this discussion, especially given discouraging past experiences [68].

At the institutional level, it's possible for the MoH to partner with UNRWA and INGOs to provide PR. Despite specific issues, these institutions clearly shape this healthcare system. UNRWA [5], and the International Committee of the Red Cross (ICRC) [4] have focused on providing the occupied territory with vital healthcare, offering flexibility, richness, and variability of services and expertise [4,5,69]. The MoH capitalises on these institutions and their assets by maintaining sovereignty, self-determination, and justice, a power mark disputed by these parties [62–64]. This complex equation can only be tackled if the MoH's leaders exercise power wisely, develop logical plans, and contain national expertise rather than embracing a factional policy. A call to sign a code of conduct that emphasises the MoH's policies and plans might also be considered.

At the professional level, we captured the non-adoption of clinical practice guidelines; HCPs did not disclose their adoption, and guidelines were not documented across the patients' records, which might indicate a documentation system defect or non-adoption of guidelines. Here, we argue that doctors and physiotherapists employ their experience in an ad hoc manner to treat their patients. Therefore, it's crucial to adopt and adhere to a clinical practice guideline [70], as noncompliance would impact care quality [71–73]. Documentation in PT is crucial to improve patient care, provide proof of business and legal records, and evidence of practice, enhance HCPs' communication, guide clinical decision-making [74], and improve quality through records audits [75]. However, documentation is not preferred by physiotherapists due to a lack of time, skills, and rewards [74,76,77], resulting in inconsistent and heading-only records [78,79]. To address these concerns, the Federated Electronic Health Records (FEHR) should be established [80], which is a nationwide system that encompasses electronic health records of variable disciplines and functionalities, gathering all sectors together to enhance efficiency, leverage telehealth, and help establish PTR [80]. Considering the professional issues, we highlighted the high awareness of HCPs and patients about the existing need for PR and PTR, and their willingness to adopt PTR. Awareness and willingness are crucial in the technology acceptance and adoption process [81,82], and should be invested in building the human resources to create the PTR programs.

The "physician physiotherapy" professional dilemma was prominent in this study. We highlighted the "physiotherapy is under-prioritised in MoH plans and practice", "Doctors underestimate the role of PT", "lack of professionals' efforts to enhance the experience and interprofessional collaboration", "lack of referral criteria to PT", "the lack of shared discharge decision", and "improper PT documentation". An intensive analysis of these themes indicated that the medical role influenced COPD management more than PT. Although doctors recognised PT, other themes indicated that this was not the actual practice. This contradiction might be attributed to the good performance of physiotherapists during the COVID-19 emergency. Spotlighting the "physician physiotherapy" professional dilemma, PT has moved from secondary healthcare to an autonomous practitioner, known as direct access, allowing physiotherapists to treat patients without a referral, which was authorised by the World Physiotherapy (WCPT) in 2011 [83]. However, the direct access system has been challenged by several factors, mainly medical dominance [84]. Notably, 58% of WCPT member organisations had direct access until 2012, especially in the private sector, with policymakers' and physicians' restrictions [85], and several countries still consider direct access [86–88]. Direct access is cost-effective [89], owing to its benefits of reducing time, cost, and resources [88] and improving disability and QoL [90]. In the current MoH practice, doctors tell physiotherapists what and how to treat their patients; "say cough." Physiotherapists complain of being marginalised by both the MoH and doctors: "doctors will not provide space for physiotherapists to work." Regular referral-based PT has limited physiotherapists' clinical reasoning, decision-making, and evidence-based practice, resulting in considerable irresponsibility and frustration [84], which was revealed in "physiotherapists' limited efforts to enhance their experience or collaboration with doctors." The development

of the PT profession and direct access has been challenged by a lack of professional regulations, medical dominance in the hierarchical structure, and inadequate education standards to meet competency requirements [84]. Efforts to upscale professional development and autonomy in the current MoH practice have always been challenged by the MoH's medical hierarchy and the lack of professional regulations and clinical practice guidelines. Direct access is possible, and doctors-physiotherapists partnering could build a suitable paradigm [91]. Professional regulation and clinical practice guidelines are crucial for interprofessional collaboration, professional boundaries, evidence-based practice and sound professional judgment.

The CST illustrates the national, institutional, and professional challenges to PTR development in a conflict and low-resourced area. Some challenges are common with other LMICs [92,93], while others are exclusive, making the CST crucial. The CST would help policymakers, researchers, and HCPs bridge healthcare shortages. However, political and socioeconomic factors are beyond their control. Considering the CST and the WHO Commission on the SDH to promote health equity [66], we argue that limiting the action to "evaluate the problem and the required action" would never be satisfactory for promoting health equity and launching the PTR in the target area. Therefore, the international community must isolate the healthcare sector from political conflicts, call on international institutions to work for healthcare equity, and, above all, ensure the warring parties uphold their obligations under international humanitarian law [6].

This study explored the possibility of launching PTR in an area of conflict and low resources, considering these exclusive settings, particularly the impacts of conflict. We argue that the inconsistent healthcare quality globally would make it harder to launch TR in LMICs [94]. Whereas HICs prioritise high technologies, data privacy, safety, usability, feasibility, affordability, and patient preferences [95], LMICs prioritise TR infrastructures [96] and health and digital health literacy [94]. In the current MoH policies and practice, the conflict and violence worsened the situation, making it crucial to question whether the right to "consistent quality of healthcare worldwide" would leverage building PTR [66,97,98] or whether people in conflict areas who struggle to get medications would be criticised for benefiting from a demanding PTR.

### Future research

This work lacked the PTR program specifications. It would be advisable to develop, evaluate, and compare the acceptance, engagement, and effectiveness of variable PTR programs in the target areas. And explore other conflict areas for further analysis.

### Strengths of the study

This study analyses the global and national shortages of the COPD management phenomena. We expect policymakers, researchers, and HCPs to use this work to reform the healthcare system, fill the shortages, build plans, initiate PTR, and provide the best possible quality of care.

### Limitations of the study

The IDI was conducted in Arabic to allow open discussion. None of the data analysis software could handle Arabic transcripts, so we had to translate and verify them into English, which consumed time and effort and complicated the qualitative data analysis.

### Conclusion

This study addressed the challenges and opportunities of launching PTR in areas of conflict and low resources, like the occupied territory. The primary issues preventing service needs and TR's establishment were national, institutional, and professional issues, such as the nonpriority in MoH practice and the fragmented COPD patient management. The MoH might cooperate with NGOs and alter its policies to provide PTR services. Additionally, interprofessional collaboration may

be needed to implement PTR. Supporting health equity necessitates the international community and INGOs to put aside the healthcare sector from conflicts and apply more efficient strategies. The intended PTR program will consist of mHealth and therapeutic devices, such as OPEP, and will start with a hybrid regime and be supported by professional associations guidelines.

## Supporting information

**S1 Table. Results of document analysis and SD questionnaire.**
(DOCX)

**S2 Table. Patients' experience with in-hospital PT, need for PR, and willingness to adopt PTR.**
(DOCX)

**S3 Table. Thematic framework 1–3: Challenges.**
(DOCX)

**S4 Table. Thematic framework 2–3: Opportunities.**
(DOCX)

**S5 Table. Thematic framework 3−3: Characteristics of the intended PTR.**
(DOCX)

## Acknowledgments

The authors would like to acknowledge all study participants for their contributions. We thank the managers at the MoH for enabling the organisation of IDIs and all colleagues for facilitating the discussions and data collection.

## Author contributions

**Conceptualization:** Suad J. Ghaben, Badr Elkholi.

**Data curation:** Badr Elkholi, Reem Kullab, Majd Al-Hour.

**Formal analysis:** Suad J. Ghaben, Badr Elkholi, Reem Kullab, Majd Al-Hour.

**Investigation:** Suad J. Ghaben, Badr Elkholi.

**Methodology:** Suad J. Ghaben, Badr Elkholi, Reem Kullab, Majd Al-Hour.

**Project administration:** Arimi Fitri Mat Ludin, Devinder Kaur Ajit Singh.

**Resources:** Suad J. Ghaben.

**Software:** Suad J. Ghaben.

**Supervision:** Suad J. Ghaben, Arimi Fitri Mat Ludin, Devinder Kaur Ajit Singh.

**Validation:** Suad J. Ghaben, Arimi Fitri Mat Ludin, Devinder Kaur Ajit Singh.

**Visualization:** Suad J. Ghaben, Arimi Fitri Mat Ludin, Devinder Kaur Ajit Singh.

**Writing – original draft:** Suad J. Ghaben.

**Writing – review & editing:** Suad J. Ghaben, Arimi Fitri Mat Ludin, Devinder Kaur Ajit Singh.

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
