## [Decision Letter · Decision Letter 0]

2 Jan 2025

PONE-D-24-30276Health Inequity; Possibilities of Initiating Pulmonary Telerehabilitation Programs for Adults with Chronic Obstructive Pulmonary Disorders in Conflict and Low-Resourced Areas; A mixed-methods Phenomenological StudyPLOS ONE

Dear Dr. Ghaben,

Thank you for submitting your manuscript to PLOS ONE. After careful consideration, we feel that it has merit but does not fully meet PLOS ONE’s publication criteria as it currently stands. Therefore, we invite you to submit a revised version of the manuscript that addresses the points raised during the review process.

We look forward to receiving your revised manuscript.

Kind regards,

Diphale Joyce Mothabeng, PhD

Academic Editor

PLOS ONE

**Journal Requirements:**

2. We note that this data set consists of interview transcripts. Can you please confirm that all participants gave consent for interview transcript to be published?

If they DID provide consent for these transcripts to be published, please also confirm that the transcripts do not contain any potentially identifying information (or let us know if the participants consented to having their personal details published and made publicly available). We consider the following details to be identifying information:

- Names, nicknames, and initials

- Age more specific than round numbers

- GPS coordinates, physical addresses, IP addresses, email addresses

- Information in small sample sizes (e.g. 40 students from X class in X year at X university)

- Specific dates (e.g. visit dates, interview dates)

- ID numbers

Or, if the participants DID NOT provide consent for these transcripts to be published:

- Provide a de-identified version of the data or excerpts of interview responses

- Provide information regarding how these transcripts can be accessed by researchers who meet the criteria for access to confidential data, including:

a) the grounds for restriction

b) the name of the ethics committee, Institutional Review Board, or third-party organization that is imposing sharing restrictions on the data

c) a non-author, institutional point of contact that is able to field data access queries, in the interest of maintaining long-term data accessibility.

d) Any relevant data set names, URLs, DOIs, etc. that an independent researcher would need in order to request your minimal data set.

For further information on sharing data that contains sensitive participant information, please see: https://journals.plos.org/plosone/s/data-availability#loc-human-research-participant-data-and-other-sensitive-data

If there are ethical, legal, or third-party restrictions upon your dataset, you must provide all of the following details (https://journals.plos.org/plosone/s/data-availability#loc-acceptable-data-access-restrictions):

a) A complete description of the dataset

b) The nature of the restrictions upon the data (ethical, legal, or owned by a third party) and the reasoning behind them

c) The full name of the body imposing the restrictions upon your dataset (ethics committee, institution, data access committee, etc)

d) If the data are owned by a third party, confirmation of whether the authors received any special privileges in accessing the data that other researchers would not have

e) Direct, non-author contact information (preferably email) for the body imposing the restrictions upon the data, to which data access requests can be sent

**Additional Editor Comments:**

Thank you for the submission. The journal will revert back to you regarding further actions.

Reviewers' comments:

Reviewer's Responses to Questions

**Comments to the Author**

1. Is the manuscript technically sound, and do the data support the conclusions?

Reviewer #1: Partly

Reviewer #2: Yes

2. Has the statistical analysis been performed appropriately and rigorously? 

Reviewer #1: I Don't Know

Reviewer #2: No

3. Have the authors made all data underlying the findings in their manuscript fully available?

Reviewer #1: No

Reviewer #2: Yes

4. Is the manuscript presented in an intelligible fashion and written in standard English?

Reviewer #1: No

Reviewer #2: Yes

5. Review Comments to the Author

**Reviewer #1: ** Abstract:

The abstract should be revised for grammar and language editing. The aim in the abstract is not clear. Methods not clear with regards to what has been investigated / determined. What was the exact conclusion of the study?

Introduction:

Line 52 – To which conflict are you referring or is this in general? Add the latest data for COPD cases and not only 2019. List examples of environmental factors limiting the access to Pulmonary rehabilitation programs. To whom is the “others” in line 72 referring to? In general, the introduction should be revised to keep thoughts together and in a logical flow. In the current format the facts are not following each other. What will the Pulmonary Telerehabilitation program consists of? How will communication be confirmed in a conflict area in order for telecommunication?

Methodology:

The methodology is not repeatable. Study process and data collection tools used are not clear. The aim mentioned in line 105 and 110 are not aligned. To whom was the questionnaire given? Who was part of the interviews? How will this research inform the Smart-OPEP development? Who were included for the sampling? How were participants recruited? Survey and questionnaire are used interchangeably – be specific in which tool was used for which objective.

Results:

Suggest present the result according to the objectives. Data tables not visible in the document.

Discussion & Conclusion:

The author needs to justify and interpret their findings and compare with other literature. The novelty of the study is not clear in the discussion.

General:

Suggestion that the manuscript be submitted for language and grammar editing. The manuscript should be written in scientific language and be more concise.

**Reviewer #2: ** Thankyou for the opportunity to review this informative work. I would like to commend the authors for embarking in this topic and aligning with transformation in healthcare in the fourth industrial revolution where telehealth is important.

Below I will cite my suggestions on areas when emphasis needs to be made:

1. Abstract: Please add the following: The journal starts with objective not background. Please include how qualitative data was analysed. Add key word= low resourced environment.

2. The Background is written well, I have a few comments which are editorial.

3. Methods: Please clarify if you conducted a retrospective records review or document analysis, if document analysis-provide the framework that guided the document analysis. Clarify what did the questionnaire measure, how many sections did it have? And describe its development before it was administered to participants. Please mention the language/s used when conducting interviews. How many days post hospital discharge were considered for patients to be included in the study.

4. Data Analysis: Quantitative data analysis is described well; however qualitative data analysis is not detailed. Qualitative phase analysis needs to be clarified. COREQ is a checklist for reporting qualitative research and not for analysing data. Please clarify which methods was used to get themes.

5. Results: Triangulation of results is well described and the three themes explain the quantitative results.

6. Discussion and conclusion: is written well.

General:

This paper is written well, just minor editing, consistency in the use of symbols and terms. It will add value to the body of knowledge of healthcare for patients with COPD and has a potential to improve the quality of care.

Authors need to make minor corrections which involve clarifying the methodology in terms of how the questionnaire was developed? Language used and data analysis for qualitative phase. Clarify more the number of days after hospital discharge, after which participants were recruited and included in the study.

Thankyou

6. PLOS authors have the option to publish the peer review history of their article (what does this mean? ). If published, this will include your full peer review and any attached files.

**Do you want your identity to be public for this peer review?** For information about this choice, including consent withdrawal, please see our Privacy Policy .

Reviewer #1: No

Reviewer #2: **Yes: ** Prof. Nombeko Mshunqane

---

## [Author Response · Author response to Decision Letter 1]

18 Apr 2025

I have uploaded a "response to reviewers" file to the system and attached a proofreading certificate. The file responds to all comments from the editor and both reviewers.

---

## [Editor Report · Decision Letter 1]

29 Apr 2025

Health Inequity: Possibilities of Initiating Pulmonary Telerehabilitation Programs for Adults with Chronic Obstructive Pulmonary Disorders in Conflict and Low-Resourced Areas; A Mixed-method Phenomenological Study

PONE-D-24-30276R1

Dear Dr. Ghaben,

We’re pleased to inform you that your manuscript has been judged scientifically suitable for publication and will be formally accepted for publication once it meets all outstanding technical requirements.

Kind regards,

Diphale Joyce Mothabeng, PhD

Academic Editor

PLOS ONE

Additional Editor Comments (optional):

Thank you for responding to the reviewer comments. The journal will revert back to you with further instructions.
---

## [Editor Report · Acceptance letter]

PONE-D-24-30276R1

PLOS ONE

Dear Dr. Ghaben,

I'm pleased to inform you that your manuscript has been deemed suitable for publication in PLOS ONE. Congratulations! Your manuscript is now being handed over to our production team.

Kind regards,

on behalf of

Dr. Diphale Joyce Mothabeng

Academic Editor

PLOS ONE